# Direct Pulp Capping: Which is the Most Effective Biomaterial? A Retrospective Clinical Study

**DOI:** 10.3390/ma12203382

**Published:** 2019-10-16

**Authors:** Anabela Paula, Eunice Carrilho, Mafalda Laranjo, Ana M. Abrantes, João Casalta-Lopes, Maria Filomena Botelho, Carlos Miguel Marto, Manuel M. Ferreira

**Affiliations:** 1Institute of Integrated Clinical Practice, Institute for Clinical and Biomedical Research (iCBR), Centre for Innovative Biomedicine and Biotechnology (CIBB), Area of Environment Genetics and Oncobiology (CIMAGO), CNC.IBILI, Faculty of Medicine, University of Coimbra, 3000-075 Coimbra, Portugal; eunicecarrilho@gmail.com; 2Biophysics Institute, Institute for Clinical and Biomedical Research (iCBR), Centre for Innovative Biomedicine and Biotechnology (CIBB), Area of Environment Genetics and Oncobiology (CIMAGO), CNC.IBILI Consortium, Faculty of Medicine, University of Coimbra, 3000-548 Coimbra, Portugal; mafaldalaranjo@gmail.com (M.L.); margaridaabrantes@gmail.com (A.M.A.); joao.casalta@gmail.com (J.C.-L.); mfbotelho@fmed.uc.pt (M.F.B.); 3Radiation Oncology Department, Coimbra University Hospital Centre, 3000-548 Coimbra, Portugal; 4Institute of Experimental Pathology, Institute for Clinical and Biomedical Research (iCBR), Centre for Innovative Biomedicine and Biotechnology (CIBB), Area of Environment Genetics and Oncobiology (CIMAGO), CNC.IBILI, Faculty of Medicine, University of Coimbra, 3000-548 Coimbra, Portugal; mig-marto@hotmail.com; 5Institute of Endodontics, Institute for Clinical and Biomedical Research (iCBR), Centre for Innovative Biomedicine and Biotechnology (CIBB), Area of Environment Genetics and Oncobiology (CIMAGO), CNC.IBILI, Faculty of Medicine, University of Coimbra, 3000-075 Coimbra, Portugal; m.mferreira@netcabo.pt

**Keywords:** biocompatibility, biomaterials, clinical, direct pulp capping, pulp vitality

## Abstract

(1) Background: Recently, tricalcium silicate cements, such as Biodentine™, have emerged. This biomaterial has a calcium hydroxide base and characteristics like mineral aggregate trioxide cements, but has tightening times that are substantially more suitable for their application and other clinical advantages. (2) Methods: A retrospective clinical study was conducted with 20 patients, which included a clinical evaluation of the presence or absence of pulp inflammation compatible symptoms, radiographic evaluation of the periapical tissues, and structural alterations of the coronary restoration that supports pulp capping therapies with Biodentine™ and WhiteProRoot^®^MTA. (3) Results: This clinical study revealed similar success rates between mineral trioxide cement and tricalcium silicates cements at 6 months, with 100% and 95% success rates, respectively. There were no statistically significant differences between both biomaterials and between these and the various clinical circumstances, namely the absolute isolation of the operating field, exposure size, the aetiology of exposure, and even the type of restorative material used. (4) Conclusions: Biodentine™ demonstrated a therapeutic effect on the formation of a dentin bridge accompanied by slight inflammatory signs, with a high clinical success rate, indicating the possibility of its effective and safe use in dental pulp direct capping in humans, similar to the gold standard material.

## 1. Introduction

Pulp protection therapies aim to maintain the vitality of the tooth in which pulp tissue has been exposed due to trauma, carious lesions, or restorative procedures. Treatment options in this situation include direct pulp capping, pulpotomy, or pulpectomy [1,2].

Traditionally, direct pulp capping therapies were performed on young permanent teeth with iatrogenic pulp exposures or trauma. When pulpal exposures occurred after carious tissue removal, in most situations, the recommendation was to perform pulpotomies or even pulpetomies. Direct pulp capping was restricted to very specific cases with very narrow indications. Many materials have been used for this type of therapy, as Pereira Paula et al. described [3]. Although calcium hydroxide-based cements have been the material of choice for many years, it presents some disadvantages, as has been widely reported: poor adherence to dentin, dissolution over time, and multiple tunnel defects in the formed dentin bridges [1]. Further, its success rates ranged from 30% to 85% [4,5,6]. With the development of other biomaterials, aggregate-based cements of mineral trioxides have emerged, which present higher success rates, form structurally more consistent dentin bridges, and with high antibacterial effects, they are now considered the gold standard materials for this type of therapy [1,7,8,9,10]. Mineral trioxide aggregate-based cements are bioactive materials that can create an environment that enhances tissue regeneration and healing. When applied in direct contact with tissues, it forms calcium hydroxide, which releases calcium ions, of which are fundamental for cell adhesion and proliferation. Additionally, it creates an antibacterial environment by increasing the pH (alkalizing the medium), modulates cytokine production, induces differentiation and migration of cells that are capable of producing dentin or bone tissue, and forms hydroxyapatite or carbonate apatite on its surface, inducing a biological sealing [11]. Hydroxyapatite can release calcium and phosphorus ions continuously, contributing to hard tissue metabolism, marginal sealing with dentine tissue, and hard tissue regeneration and remineralization [12,13,14]. MTA (mineral trioxide aggregate) also has the lowest cytotoxicity values [15]. However, MTA-based cements present great difficulties regarding clinical manipulation, high costs, and long setting times [1,16]. 

The emergence of new bioceramic materials, with properties similar to MTA-based cements and with improved handling characteristics, suggests the possibility of a clinical alternative [9,16,17,18,19]. Tricalcium silicate is bioactive and, when hydrated, turns into hydrated calcium silicate and calcium hydroxide. These react in the presence of physiological fluids, producing hydroxyapatite, which is biocompatible and induces pulp cell differentiation in the same way as calcium hydroxide [20].

The present retrospective preliminary clinical study aims to address clinical success/failure factors following the treatment of direct pulpal exposure with WhiteProRoot^®^MTA or Biodentine™. The evaluation of these factors included the clinical evaluation of the absence or presence of symptomatology that are compatible with inflammatory pulp states and radiographic evaluation of the structural alterations of coronary restoration that support pulp tissue repair and changes in the pulp tissue itself. Overall, we sought to evaluate the advantage of the clinical use of Biodentine™ compared to other similar materials, such as WhiteProRoot^®^MTA, by searching for clinical signs of pulp tissue vitality preservation in patients who were treated with this material.

## 2. Materials and Methods 

The retrospective clinical study was approved by the FMUC/CHUC Ethics Committee with number 005-CE-2014.

### 2.1. Sample Selection

The medical files of patients who underwent direct pulp capping therapies at the Operative Dentistry appointment of the Faculty of Medicine, University of Coimbra and Coimbra University’ Hospital Centre, Portugal between 2012 and 2016 were evaluated. Only patients with direct pulp cappings with two dentinogenesis-inducing materials, a mineral trioxide aggregate, WhiteProRoot^®^MTA (Dentsply, Maillefer, Tulsa Dental Specialties, Switzerland), and a tricalcium silicate, Biodentine™ (Septodont, Saint-Maur-des-Fosses, France), were included. The goal was to standardize the sample, with a similar number of clinical cases for each of the biomaterials.

### 2.2. Inclusion Criteria

The included patients met certain criteria which were reported in the clinical chart, namely: thermal sensitivity response compatible with vital tooth diagnosis and corroboration by radiographic examination; patients aged 18 to 55; and a reasonable state of health and oral hygiene, without periodontal pathology. The included treatments were made on absolute isolation with a rubber dam or with a relative isolation of the operative field with cotton rolls. Cavity disinfection was performed with 2% chlorhexidine. Teeth in which restorations were performed using either total-etch adhesive or self-etching adhesive systems were included. The restorative materials that were used in all the therapies were light-cured composite resins.

### 2.3. Evaluation of the Clinical Procedures of Direct Pulp Capping Therapies

Twenty-one direct pulp capping therapies performed on 20 patients were included, divided into two groups—one with 11 teeth that were treated with WhiteProRoot^®^ MTA and the other with 10 teeth that were treated with Biodentine™. The aetiologies of the pulp exposures were different, namely due to extensive carious lesion and iatrogenic causes and, therefore, different dimensions of the exposure sites were reported. Some clinical files also referred to the need for pulp cappings before the complete removal of the active caries lesion. In all of the selected patients, the therapies were performed with one of the biomaterials, followed by the placement of a resin-modified glass ionomer cement. The definitive restorations were performed at the appointment in which the direct pulp capping was performed or, in some cases due to various constraints, within 15 days after. At the end of each therapy, the radiographic evaluation was reported. All these clinical procedures were performed following the standards of good clinical practice [21].

### 2.4. Evaluation of the Teeth after Therapy

Clinical controls were performed at one week, one month, three months, and six months after treatment. Broader follow-ups could not be obtained because no clinical record of further evaluations was found after 6 months. In these controls, qualitative tests had been performed such as the pulp sensitivity test, which included the ethyl chloride cold test and the vertical and horizontal percussion test. The criteria that were used in the qualitative assessment have already been established as good practice procedures and are described in Table 1. Clinical controls over time assessed the restoration success, including the search for fractures and/or marginal infiltration, through clinical examination and complementary exams with periapical radiography. This radiographic examination also allowed the evaluation of secondary signs of treatment failure, namely the formation of apical root injury and the appearance of secondary caries. Therapies on teeth that remained asymptomatic, with normal ethyl chloride sensitivity tests and no radiographic signs of periapical pathology, were considered as clinical successes.

### 2.5. Statistical Analysis

The age factor was evaluated through descriptive analysis and the application of normality tests, namely the Kolmogorov-Smirnov and Shapiro-Wilk tests. For comparisons of the biomaterials that were used in the therapies and pulp exposure characteristics, namely aetiology and size, the chi-square test or Fisher’s exact test according to Cochrane rules were used. To evaluate the variables that characterize the symptoms over time, a frequency analysis was performed.

## 3. Results

The distribution of the clinical parameters that were considered can be seen in Figure 1. The present clinical study included patients aged 18 to 55 years, most of which were aged 18 to 25 years, with a percentage of 45%. The age group of 46 to 55 years was the second-highest group to be represented, with a percentage of 30% of the total patients. The mean age was 32.2 ± 13.85 years, with a non-normal distribution, a median of 26, and an interquartile range (AiQ) of 28.

Regarding the type of teeth with pulp capping therapy, the distribution was uniform with a higher percentage of molars, 42.9%, followed by the group of incisors and canines with 33.3% and, finally, premolars with 23.8%. Hygiene habits were mostly moderate and good, with a percentage of 42.9% in both groups. Only a small percentage of 14.3% was referred to as having poor hygiene habits.

The biomaterials that were used to protect the pulp tissue and the adhesive and restorative materials that were used in the restorative phase can be seen in Figure 2.

WhiteProRoot^®^MTA was used in 52.4% of the teeth and Biodentine™ in 47.6%. Regarding restorative materials, about 23.8% of the teeth were initially restored provisionally with a glass ionomer cement and about 76.2% were permanently restored. In the permanent restoration of all the teeth that were submitted to pulp capping therapy, a total-etch dentin bonding system was used in 57.2% and a self-etch dentin bonding system was used in 42.8%. The definitive restorations were made with composite resins, as already mentioned.

Relevant pre-therapeutic diagnostic factors and those that occurred during therapy and were relevant for prognosis were also recorded, as shown in Figure 3.

Regarding pre-therapeutic diagnosis factors, pulp exposure in about 85.7% of the intervened teeth was caused by dental caries. On the other hand, about 14.3% of pulp exposure was iatrogenic. No pulp exposure due to dental trauma was reported. For the therapies that were applied to the 21 teeth, about 95.2% were performed under rubber dam isolation and in the others, relative insulation with cotton rolls was used. In 14.3% of the therapies that were reported, there was decayed dentin at the time of pulp exposure. Regarding pulp exposure, its size was less than 1 mm in 47.6% of the treated teeth and an average size of 1 to 2 mm was reported in 47.6% of the teeth. Exposures that were larger than 2 mm were reported in 4.8% of the teeth. There were reports of bleeding in 71.4% of clinical situations and no bleeding was reported in 28.6% of pulp exposures.

As mentioned, data from the initial clinical evaluation were extracted from all the intervention teeth to confirm the diagnosis of vital or reversible pulpitis, compatible with the direct pulp capping therapy that was instituted as the treatment. These initial assessment data are described in Figure 4.

Thus, in all teeth undergoing therapy, the reported horizontal and vertical percussion tests were negative. Sensitivity testing with ethyl chloride revealed a normal response in 95.2% of the teeth and slightly increased, i.e., was compatible with reversible pulpit tooth, in 4.8% of the teeth.

Evaluations were performed at 1 week, 1 month, 3 months, and 6 months after treatments, considering the various parameters to measure the pulp state of the teeth that were intervened. In all the controls, a radiographic evaluation was performed, which contributed to the evaluation of some of these parameters. Figure 5 shows periapical radiographs, by way of example, from two clinical cases in which pulp capping with different biomaterials was made.

The post-treatment evaluations considered five parameters and the results can be seen in Figure 6. In evaluations that were performed in the first-week control, the sensitivity test was normal in 71.4% of teeth and slightly increased in 28.6% of teeth. In the horizontal percussion test, it was reported that 95.2% of the teeth responded negatively and 4.8% responded positively (reported for the Biodentine^®^ group). Regarding the vertical percussion test, it was reported that 90.5% of the teeth presented negative answers and 9.5% reported positive answers (reported to the Biodentine^®^ group and the WhiteProRoot^®^MTA group by an equal percentage). Radiographic evaluation revealed that 100% of the intervened teeth showed no signs of secondary caries, nor any signs of apical tissue injury.

At 1 month, evaluation sensitivity testing was normal in 65% of teeth, slightly increased in 30% of teeth, and greatly increased in 5% of teeth. On the other hand, in all subjects undergoing therapy, the horizontal and vertical percussion tests were negative. Radiographic evaluation revealed that 100% of the intervened teeth showed no signs of secondary caries, nor any signs of apical tissue injury.

In the 3 months assessment, the percentage of teeth with a normal sensitivity test increased to 80%. A slight increase in sensitivity was revealed in about 20% of the teeth. The reported vertical percussion tests were negative in 100% of the teeth, as opposed to the horizontal percussion test, which was positive in 5%. As in the 1 week and 1 month controls, complementary radiographic examinations revealed that 100% of the intervened teeth showed no signs of secondary caries or injury to the apical tissues.

The post-therapeutic evaluation at 6 months revealed some changes from the other control times. Thus, pulp necrosis was diagnosed in one tooth (5%) and a marked increase in sensitivity was diagnosed in another (5%), both of which reported to the Biodentine^®^ group. Sensitivity testing was normal for 90% of the teeth that were evaluated. Corroborating the evaluation of the sensitivity tests, the vertical percussion test was positive in one tooth (5%) and negative for the remaining 95% of the teeth. The horizontal percussion test was negative in 100% of the teeth. Radiographic evaluation, both at the coronary level and at the root level, retained negative results regarding the appearance of secondary caries and only had signs of periapical tissue damage in one tooth (5%), which was reported to the Biodentine^®^ group as referred.

In addition to this descriptive analysis, comparisons were also made between the results that were obtained. Thus, comparisons were made between the clinical situations that were presented before therapy and the results that were obtained after therapy. Some patient data, including age, were compared with the results of the sensitivity tests over time, and hygiene habits were compared with the results of the radiographic assessment of coronary restoration over time. These comparisons showed no statistically significant differences. Comparisons were also made between the circumstances of pulp tissue exposure (caries or iatrogenic), the cavity situation at the time of exposure (with or without caries), and the size of the exposure, with the results of sensitivity tests over time. From these comparisons, it was concluded that there were no significant differences between the various circumstances and the results of these tests.

Finally, comparisons were also made between the two biomaterials that were studied and the results of the sensitivity tests with ethyl chloride and the horizontal and vertical percussion and the radiographic evaluation radicular. It was concluded that there were no statistically significant differences between the biomaterials and the results that were obtained after therapy.

Both biomaterials had similar success rates at 6 months, i.e., 100% for WhiteProRoot^®^MTA and 95% for Biodentine™. In this last group, there was a clinical situation of necrosis.

## 4. Discussion

Whenever possible, this study sought to rigorously apply the methodological items for non-randomized studies, MINORS version, to increase its reproducibility and quality. Since this is a retrospective study, the inclusion criteria were applied in the search and selection of the clinical files. Retrospective data were collected from each clinical process, at the endpoints that were determined in the study’s objective, without biased evaluation. No prospective sample size calculation was performed as this is a preliminary study. Considering published meta-analysis studies, which revealed very similar clinical performances between MTA-based cement and tricalcium silicate-based cement, and in order to clinically evaluate the efficacy of these therapies, only two biomaterials were chosen [3,22,23]. Thus, with such restrictive criteria in the choice of biomaterial, it was not possible to obtain a large sample. Additional criteria for comparative studies were applied, with the choice of the gold standard biomaterial as the control group, the contemporaneity of the biomaterials, the base equivalence in both groups, and appropriate statistical analysis.

The results that were observed in the present study are comparable with others, showing success rates of 92.5% for Biodentine™ and 84.6% for WhiteProRoot^®^MTA [2]. Other studies mention success rates of 85%, 91.7%, and 97.1% for MTA-based cements after 3, 1, and 9 years of follow up, respectively; and 83.3% and 85.37% for 1-year follow-up with tricalcium silicate cements [1,10,24,25,26,27,28]. Some authors report similar success rates between the two types of biomaterials in immature permanent teeth [29,30,31].

The treatment failure is referred to in some studies as occurring in the first weeks after therapy, mainly due to previous impairment of the pulp state [2]. This is difficult to diagnose through sensitivity tests as these tests do not assess pulp vitality, but only measure sensitivity through subjective symptomatic assessment of the patient. In the present study, the only treatment failure occurred after 6 months, which was evaluated with the positive vertical percussion test and negative ethyl chloride test, inferring that pre-therapeutic diagnoses were correctly made. An individual cause is likely to be related to this loss of vitality since throughout the other assessment times, the tooth never changed in sensitivity testing and radiographic assessment. However, the small sample size of the present study may have contributed to the absence of statistically significant differences in the success rates of the different groups. In fact, over the various post-therapy periods, only the time and intensity of the positive response to the ethyl chloride test increased in some cases. Nevertheless, it was found that this change in response gradually decreased from week 1 to 6 months, either in the percentage of clinical cases or in response intensity. Percussion tests were positive in some cases, but sporadically and without recurrence in the various subsequent control times.

It is recognized that the direct pulp capping therapies’ success depends on the rigorous selection of cases and the appropriate application of the treatment protocol [32]. Thus, the inclusion criteria for this clinical study took this premise into account. Many of the cases that were included in the present study had the aetiology of dental caries exposure. In this case, it is difficult clinically to assess the true pulp state, which is of extreme importance for prognosis. Some authors suggest that the degree of haemorrhage after exposure is a good indicator to measure the inflammatory state of the pulp tissue [32]. The occurrence of intense bleeding, which is difficult to control, is suggestive of pulp with a moderate to severe inflammatory stage and, therefore, without indication for direct pulp capping therapy [33]. Control of this pulp haemorrhage and cavity disinfection during direct pulp capping therapies are decisive steps that can affect pulp tissue regeneration. Bleeding should be controlled by exerting pressure at the exposure site for 3 to 5 min with a sterile cotton ball soaked in the haemostatic/disinfection solution. Several authors have studied the efficacy and influence of these solutions on pulp tissue regeneration [34,35,36,37]. Although saline has limited effects, it is traditionally the most widely used. Some authors report that 0.9% of saline has acceptable results, but is inferior to other solutions such as chlorhexidine and sodium hypochlorite [35,36,37]. The established protocol recommends the use of chlorhexidine at a concentration of 2% and with a 3 min application time to achieve maximum efficacy with minimal cytotoxicity, according to the literature [35,38,39]. In all clinical processes that were consulted, chlorhexidine was the cavity disinfectant that was chosen in the direct pulp capping therapies, and the procedure was made according to the protocol.

The radiographic technique of long cone parallelism is not a practice that is established in outpatient dentistry appointments and there are no standardized radiographic records of the therapies that were performed in the clinical processes. Thus, it was not possible to perform a radiographic comparison study. This would have been useful in gauging the formation and quality of the dental bridge, as was done in other studies [31,40]. The results of those report very similar values between biomaterials, between 85% and 95%, when the evaluation refers to the existence of dentin bridge, although incomplete [24,30]. However, it is also described in the literature that the relatively low radiopacity of tricalcium silicate-based cement, when compared with MTA-based cements, can make this type of assessment difficult [2,41,42,43].

In extracting data from clinical files, it was found that the use of glass ionomer cement liner had always been performed after biomaterials were applied to the exposure sites. However, according to some authors and the manufacturer of the tricalcium silicate-based biomaterial, the setting time is very short and definitive restoration can be performed in the same appointment without the need to place a base between the biomaterial and the restorative materials. Some authors also report the use of tricalcium silicate cement as an integral substitute for all dentine tissue using a sandwich restorative technique [1,16,18,19].

The adhesive techniques that were performed in this study, whether using total-etch or self-etch adhesive systems with selective enamel etching, are recommended by many authors as having the best adhesion values and, consequently, the best performance [44,45,46,47,48,49]. It was also observed that the definitive restorations were performed by direct or indirect technique with the composite resins, as recommended by the literature [39,50,51,52,53,54]. The results of the radiographic evaluations that were performed on the restorations throughout the study that reveal the absence of signs of secondary caries corroborate this recommendation.

Histologically, it is established that mineral trioxide aggregates induce the formation of a thicker hard tissue barrier than calcium hydroxide-based materials, accompanied by less tissue inflammation. In addition, pulp tissue has progenitor cells that are capable of differentiating into odontoblast cells type, with the capacity to mineralize tissue deposition. This mechanism is facilitated by mineral trioxide aggregates by inducing the secretion of morphogenetic proteins and growth factors such as BMP-2 and TGF-β1 [55,56]. The technology behind the biosilicate manufacturing process, the main constituent of Biodentine™, removes metallic impurities that are present in other cements. The main reaction involves the hydration of tricalcium silicate and the production of a calcium silicate and calcium hydroxide based gel which in contact with phosphate ions, has the ability to precipitate a hydroxyapatite-like compound [57]. Some studies even report that due to its low cytotoxicity and high bioinductive capacity, Biodentine™ can be considered the ideal material for the treatment of pulp exposures [55]. The repair mechanism by deposition of mineralized tissue depends on the pH and the ability of tricalcium silicate cement to release various ions. Extracellular calcium ions represent a potent factor that affects cell function and has significant effects on cell proliferation and differentiation. Calcium ions are known to be involved in a number of proliferation signaling pathways and represent a key point for the investigation of the action mechanisms of these biomaterials and the eventual development of others.

The main limitations of this retrospective clinical study are the small sample size and the short evaluation period, so it was considered preliminary. Longer evaluation times and larger samples would be important to assess intrapulpar changes resulting from these therapies and the effective formation of a dentin bridge at the site of exposure [58]. It would also be advisable to histologically evaluate pulp capping therapies over time if possible. This evaluation would only be feasible in randomized clinical studies involving teeth with an indication for extraction.

## 5. Conclusions

As the use of tricalcium silicate cement is relatively new, there are few studies regarding its clinical performance in permanent teeth. In the present study, success rates of tricalcium silicate cement were 95%, which are high and similar to the 100% success rate of MTA-based cement. Comparisons showed that the success rates obtained are not dependent on age or tooth type or on the cavity or aetiology of pulp exposure. Tricalcium silicate cement could be an alternative to pulp capping with low costs, more clinically friendly setting times, and the same therapeutic performance as the gold-standard, MTA-based cement.

## Figures and Tables

**Figure 1 materials-12-03382-f001:**
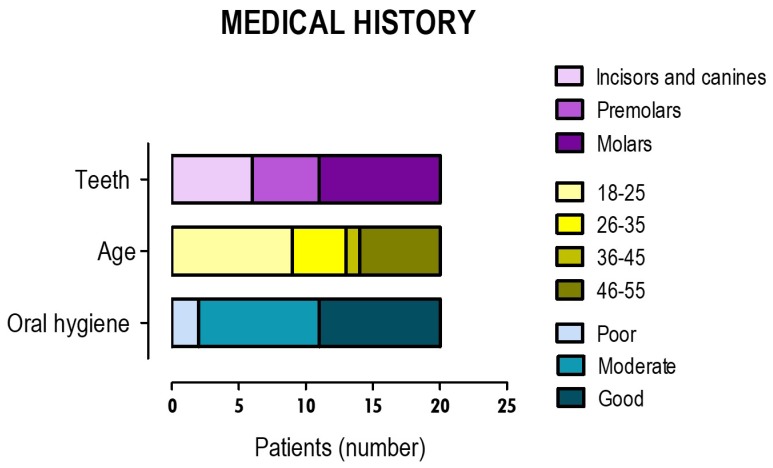
Relevant aspects of the clinical history of patients who were included in the clinical study, namely age, hygiene habits, and type of teeth intervened. Results corresponded to 20 patients in whom pulp capping therapies were performed on 21 teeth.

**Figure 2 materials-12-03382-f002:**
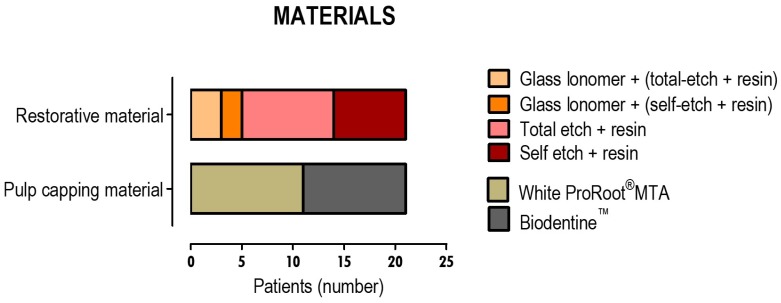
Types of material used for the direct pulp capping and restoration of the dental structure. The results are for 21 teeth, in which pulp capping therapies were performed.

**Figure 3 materials-12-03382-f003:**
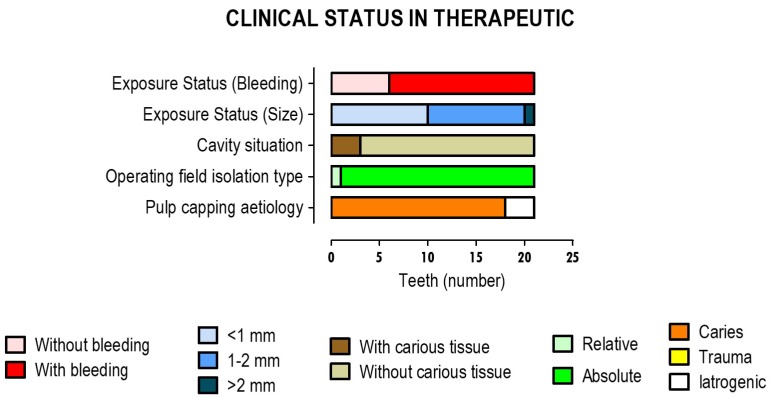
Diagnostic factors of pre-therapeutic clinical status, such as the cause of pulp capping and the type of isolation performed; and relevant clinical factors for prognosis, such as cavity status and pulp exposure status (size and bleeding). The results correspond to 21 teeth in which pulp capping therapies were performed.

**Figure 4 materials-12-03382-f004:**
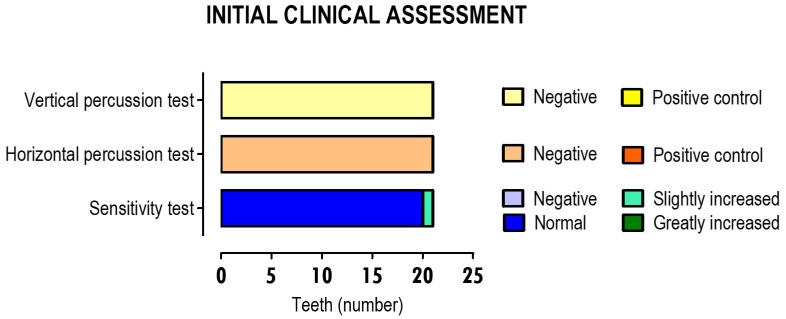
Pre-therapeutic clinical assessment, assessment of changes in cold and horizontal and vertical percussion sensitivity tests. The results correspond to 21 teeth in which pulp capping therapies were performed.

**Figure 5 materials-12-03382-f005:**
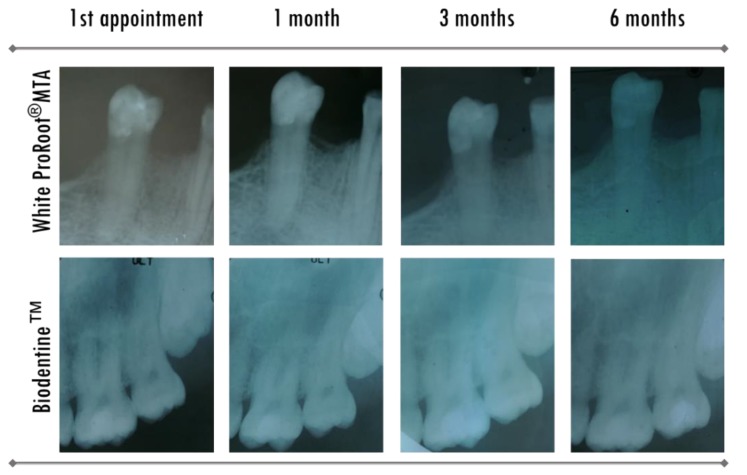
Radiographic images corresponding to direct pulp capping therapies with WhiteProRoot^®^MTA and Biodentine™ biomaterials. Therapy with WhiteProRoot^®^MTA was performed on tooth 35, with iatrogenic pulp exposure <2 mm, with haemorrhage and under absolute isolation. The definitive restoration was performed after therapy at the same appointment with a total-etch adhesive Scotchbond 1^®^ (3M/ESPE, Burgdorf, Schweiz) and composite resin Synergy D6^®^ (Coltène/Whaledent AG, Altstätten, Switzerland). Biodentine™ therapy was performed on tooth 26, with pulpal exposure due to carious lesion between 1 and 2 mm, without bleeding and under absolute isolation, and was definitively restored after therapy with a total-etch adhesive Scotchbond 1^®^ and resin composite Synergy D6^®^.

**Figure 6 materials-12-03382-f006:**
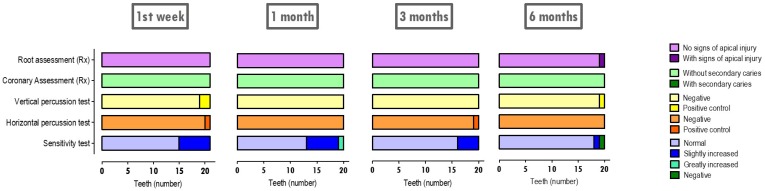
Therapeutic control times. Five parameters were analyzed: cold sensitivity tests (negative, normal, slightly increased, and greatly increased); horizontal and vertical percussion tests (negative and positive); coronary radiography (with or without secondary caries); root radiography (with or without apical lesion). The results express the evaluation of 21 teeth at 1 week and the evaluation of 20 teeth at 1 month, 3 months, and 6 months.

**Table 1 materials-12-03382-t001:** Qualitative evaluation criteria for the pulp sensitivity tests.

Test	Normal State of the Pulp Tissue	Pathological State of Pulp Tissue		
		Reversible Pulpitis	Irreversible Pulpitis	Necrosis
Test with ethyl chloride	positive (no change in intensity or duration after stimulation)	positive (with change in intensity and/or duration after lower stimulus 5 s)	positive (with change in intensity and/or duration after upper stimulus 5 s)	negative
Horizontal Percussion Test	negative	negative	positive	positive
Vertical Percussion Test	negative	negative	positive	positive

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
