# Peer review of "Direct Pulp Capping: Which is the Most Effective Biomaterial? A Retrospective Clinical Study"

_materials, 2019, doi:10.3390/ma12203382_

Round 1

Reviewer 1 Report

Dear authors
Thank you very much for this beautiful work.
The comparison of classical MTA ProRoot with Biodentine is not new as such. Very nice however are the relatively large case numbers
I would have liked to have worked out the differences between the two materials more precisely.
Which cases failed with which material? In the discussion a lot of time was spent to critically illuminate the processes of the pulp capping. Very little discussion took place about the differences between the different materials.
A more detailed analysis of the reasons for failure would be nice.
Otherwise, the study is very sound

Author Response

Reviewer 1

Dear authors
Thank you very much for this beautiful work.
The comparison of classical MTA ProRoot with Biodentine is not new as such. Very nice however are the relatively large case numbers

The authors acknowledge the reviewer’s comments.

I would have liked to have worked out the differences between the two materials more precisely.

R: The alterations were made in the manuscript.

Which cases failed with which material?

R: This information was added to the manuscript.

In the discussion, a lot of time was spent to critically illuminate the processes of the pulp capping. Very little discussion took place about the differences between the different materials.
A more detailed analysis of the reasons for failure would be nice.

R: The reviewer's suggestion was considered. Considerations on the clinical procedures of direct pulp capping were reviewed and the differences between the materials used were discussed. The alterations were made in the manuscript.

Otherwise, the study is very sound

Reviewer 2 Report

Title: Direct pulp capping: which is the most effective biomaterial? A retrospective clinical study

Interesting project. In this research project retrospective clinical study with 20 patients performed which results show supports of pulp capping therapies with Biodentine™ and WhiteProRoot®MTA.

Corrections:

English need proof reading. Never use “we” in scientific paper. Please specify where consumables purchased such as: WhiteProRoot®MTA and a tricalcium silicate, Biodentine Table 1: is this your table? If not you need to reference it. Result section – line 2: 45% does not make sense? Reword/correct the sentence:

“clinical study included patients aged 18 to 55 years, most of them aged 18 to 25 years, with a percentage of 45%. “.

Result: line 3: what do you mean by 30%? The sentence does not make sense: “The age group of 46 to 55 years was the second-highest, with a percentage of 30%.” Entire result and discussion have serious English problem and most of places mentioning % does not make sense and the readers do not get your point. You need to revise this paper. Conclusion: success rate of what? 95% of what? “In the present study, success rates were 95%, which are high and like those for 100% MTA-based cement.

Overall, interesting paper but need serious English proof reading specifically in result and discussion section. I accept after major correction.

Author Response

Reviewer 2

Interesting project. In this research project retrospective clinical study with 20 patients performed which results show supports of pulp capping therapies with Biodentine™ and WhiteProRoot®MTA.

The authors acknowledge the reviewer’s comments.

Corrections:

English need proof reading.

R: The paper was reviewed by an English native international certified revisor (Proofreading Declaration attached).

Never use “we” in scientific paper.

R: The paper was reviewed by an English native international certified revisor (Proofreading Declaration attached).

Please specify where consumables purchased such as: WhiteProRoot®MTA and a tricalcium silicate, Biodentine

R: The materials used were purchased by the hospital unit where the patients were treated. The hospital unit is part of the Portuguese health system network, where consumables are purchased from Portuguese commercial companies registered with the drug regulatory authority (Infarmed).

Some information about the materials used in this study is given in the following table.

Material

Lot

Expiration date

Trade mark

Biodentine

LOT B05574

LOT B08323

LOT B10221

LOT B12265

LOT B14676

LOT B13380

2014-04

2014-11

2015-09

2016-04

2016-07

2016-09

Septodont

Life®

LOT 4989034

LOT 4954566

LOT 4974378

LOT 5598117

2015-09

2015-09

2015-09

2017-06

Kerr

WhiteProRoot® MTA

LOT 12002493

LOT 201404-01

LOT 13102907

LOT 13102906

2015/05

2016-08

2016-11

2016-11

Denstply

Ketac Fil

LOT 496854

2015-12

3M ESPE

Corsodyl Care

LOT 000438

2013-09

GlaxoSmithKline

Sodium hypochlorite

LOT 031112

2014/11

DentaFlux

Table 1: is this your table? If not you need to reference it.

R: The table was made by the authors.

Result section – line 2: 45% does not make sense? Reword/correct the sentence:“clinical study included patients aged 18 to 55 years, most of them aged 18 to 25 years, with a percentage of 45%. “.

R: The alteration was made in the manuscript.

Result: line 3: what do you mean by 30%? The sentence does not make sense: “The age group of 46 to 55 years was the second-highest, with a percentage of 30%.”

R: 30% of total patients. The alteration was made in the manuscript.

Entire result and discussion have serious English problem and most of places mentioning % does not make sense and the readers do not get your point. You need to revise this paper.

R: The paper was reviewed by an English native.

Conclusion: success rate of what? 95% of what? “In the present study, success rates were 95%, which are high and like those for 100% MTA-based cement.

R: “… success rates of tricalcium silicate cement therapies were 95%, which are high and like those for 100% MTA-based cement.” The alteration was made in the manuscript.

Overall, interesting paper but need serious English proof reading specifically in result and discussion section. I accept after major correction.

Round 2

Reviewer 2 Report

Thanks for corrections.

All corrections done as requested/suggested. 

I accept in the current format.